# The Effect of Javanese Language Videos with a Community Based Interactive Approach Method as an Educational Instrument for Knowledge, Perception, and Adherence amongst Tuberculosis Patients

**DOI:** 10.3390/pharmacy9020086

**Published:** 2021-04-18

**Authors:** Fauna Herawati, Yuni Megawati, Retnosari Andrajati, Rika Yulia

**Affiliations:** 1Department of Clinical and Community Pharmacy, Faculty of Pharmacy, Universitas Surabaya, Jalan Raya Kalirungkut, Surabaya 60293, Indonesia; yuni.meg@gmail.com (Y.M.); rika_y@staff.ubaya.ac.id (R.Y.); 2Department of Pharmacology and Clinical Pharmacy, Faculty of Pharmacy, Universitas Indonesia, Depok 16424, Indonesia; retnosaria@gmail.com; 3Rumah Sakit Umum Daerah (RSUD) Bangil, Pasuruan 67153, Indonesia; aslichahdr@gmail.com

**Keywords:** tuberculosis, knowledge, perception, adherence

## Abstract

The long period of tuberculosis treatment causes patients to have a high risk of forgetting or stopping the medication altogether, which increases the risk of oral anti-tuberculosis drug resistance. The patient’s knowledge and perception of the disease affect the patient’s adherence to treatment. This research objective was to determine the impact of educational videos in the local language on the level of knowledge, perception, and adherence of tuberculosis patients in the Regional General Hospital (RSUD) Bangil. This quasi-experimental study design with a one-month follow-up allocated 62 respondents in the intervention group and 60 in the control group. The pre- and post-experiment levels of knowledge and perception were measured with a validated set of questions. Adherence was measured by pill counts. The results showed that the intervention increases the level of knowledge of the intervention group higher than that of the control group (*p*-value < 0.05) and remained high after one month of follow-up. The perceptions domains that changed after education using Javanese (Ngoko) language videos with the Community Based Interactive Approach (CBIA) method were the timeline, personal control, illness coherence, and emotional representations (*p*-value < 0.05). More than 95% of respondents in the intervention group take 95% of their pill compared to 58% of respondents in the control group (*p*-value < 0.05). Utilization of the local languages for design a community-based interactive approach to educate and communicate is important and effective.

## 1. Introduction

Tuberculosis (TB) is an infectious disease of international concern and is prevalent in Indonesia. In 2050, it is estimated that deaths due to anti-TB drug resistance will be 10 million more than from cancer [1]. The total global losses incurred due to anti-TB drug resistance may reach US$100 trillion. The World Health Organization (WHO) Global Report 2019 states that the total number of TB cases identified in Indonesia rose from 331,703 in 2015 to 563,879 in 2018 (+70%); a 28% increase happened between 2017 and 2018 [2]. In 2016, 110,000 people, or 42 per 100,000 population, died of TB of which 32,000 (12 per 100,000 population) were caused by Multi-Drug-Resistant Tuberculosis (MDR-TB) [3]. Pasuruan Regency is among the five cities/districts with the highest number of TB cases in East Java. In 2013, the number of TB cases in Pasuruan Regency was 964, which declined to 886 cases in 2014 and rose to 1693 cases in 2015 [4]. In the Regional General Hospital (RSUD) Bangil, there were 100 tuberculosis outpatients at the pulmonary clinic each month from January to June 2018. If the number of tuberculosis patients continues to rise, Indonesia would fail to achieve the TB control targets of the 2020–2024 National Medium-Term Development Plan (RPJMN), Millennium Development Goals (MDGs), and Sustainable Development Goals (SDGs).

Tuberculosis treatment lasts at least 6 months, depending on the clinical presentation in each patient, which causes patients to be at high risk of forgetting to take medication or dropping out of treatment. Medication non-adherence will prolong therapy duration, and increase the risk of drug resistance, morbidity, and mortality [5,6]. Resistance cannot be eliminated but can be controlled with the appropriate use of anti-TB drugs. Furthauer et al. (2013) argued that factors causing non-adherence include patients’ lack of knowledge about their health, the patient’s weak relationship with medical personnel, and the drugs’ side effects [7].

According to Regulation of the Minister of Health No. 72 of 2016, one of the roles of clinical pharmacists in controlling anti-TB drug resistance is to provide education to patients and the public about tuberculosis and the judicious use of anti-TB drugs, in the hope of increasing patients’ knowledge, which in turn shapes correct perceptions about the disease, encourages adherence and controls the number of drug resistance cases [8]. Videos in the Javanese (Ngoko) language were chosen as an instrument because a person can absorb information best and highest through sight and hearing senses; video educational material more effective than text-only [9]. A video educational tool was effective for increasing the level of outpatients’ knowledge [10,11] and remained significant after three months [10]. The study was conducted at the Regional General Hospital (RSUD) Bangil because the hospital is a referral hospital in Pasuruan Regency with a high number of TB cases. Secondary data from Patient and Family Education in the pulmonary clinic of the Regional General Hospital (RSUD) Bangil in October 2017 showed educational achievements by the health personnel were not yet optimal. This was confirmed by the pharmacist at the outpatient pharmacy, who reported that many tuberculosis patients who were following treatments at the Regional General Hospital (RSUD) Bangil did not adhere to the scheduled patients’ routine visits and no record was written on the pharmacy’s education register. This study aimed to assess the impact of a local language educational video on the level of knowledge, perception, and adherence of tuberculosis patients in the Regional General Hospital (RSUD) Bangil.

## 2. Materials and Methods

Before the Javanese (Ngoko) videos were created, the researcher performed a needs assessment and education plan for respondents so that contents could be suited to the needs of tuberculosis outpatients in the Regional General Hospital (RSUD) Bangil. The Javanese (Ngoko) language was adopted because the majority of patients use the Javanese (Ngoko) language daily. The design of this research was quasi-experimental with a control group and an intervention group. The control group and the intervention group were followed for 30 days. Data collection began by screening the medical records of prospective respondents. Prospective respondents who fulfilled the inclusion and exclusion criteria were visited, and the researcher explained the purpose of the study along with giving an informed consent form to be signed by the respondents as evidence of volunteerism. Respondents who were willing to take part in the study were allocated into a control group and intervention group with a simple random sample using a lottery method. Afterward, on day-1 and 30, respondents were given questionnaires to test their levels of knowledge and perception (Figure 1). Questions on knowledge level were adapted from several studies [12,13,14,15], guideline published by Ministry of Health Indonesia [16,17] and WHO [18]. The expected achievements on knowledge level were based on Bloom’s Revised Cognitive Domain [19,20], which were knowing, understanding, and applying. Questions on perception were adapted from The Revised Illness Perception Questionnaire [21]. The expected result was a change from negative perception to positive perception. The same questions (Appendix A) were given twice to the control group on day 1 and day 30, and three times to the intervention group on day 1 (before and after being provided with education) and day 30. The researcher performed tests on knowledge and perception during patients’ routine visits at Bangil District Public Hospital. An educational video in the local language about tuberculosis disease, anti-TB drugs administration, and their adverse drug events was given to educate the community (TB outpatients) in a small discussion group, named Community-Based Interactive Approach (CBIA), at the pulmonary clinic of the Regional General Hospital (RSUD) Bangil.

### 2.1. Respondents

Data collection for the study sample was performed from October to December 2018 at the pulmonary clinic of the Regional General Hospital (RSUD) Bangil, Pasuruan Regency (Figure 1). The recruitment flow of TB respondents can be seen in Figure 2. Inclusion criteria were tuberculosis patients >14 years who received anti-TB drug category one and two, while exclusion criteria were tuberculosis patients who were currently following the Directly Observed Treatment Short-Course (DOTS) program, patients diagnosed with schizophrenia, blindness, or deafness.

Education was given by the researcher to the intervention group through videos in Javanese (Ngoko) language with a CBIA approach upon the completion of their clinic visit on day 1. The educational video (CBIA) duration was six minutes. It covered information about tuberculosis disease, tuberculosis treatment (including duration, the risk of drug resistance, and adverse drug events), a reminder system, and non-pharmacology aspects. The control group received standard care, education on drug administration from health care. CBIA was done through small group discussions between 6 and 8 respondents; each group was accompanied by a counselor who facilitated the discussions. Respondents were encouraged to be more active in expressing opinions and asking questions of the informant about the discussion material, and the outcome of these small group discussions was ultimately presented to all groups. During the implementation, every CBIA education session (the control group and the intervention group), was assisted by a group of 3–4 people, comprising of a physician, pharmacist, pharmacy student, and/or medical student who had previously been briefed. To anticipate respondents forgetting the educational material, each respondent was provided with videos on his or her mobile phone. The videos were transferred from the researcher to the respondent’s mobile devices (with Bluetooth, Share It, WhatsApp, or LINE platform).

To understand the effect of education on adherence, adherence to treatment in the control group and intervention group was measured by pill counts on day-1 and 30; the number of drugs taken by the patients with counting the remaining units (drugs consumed) divided by the number of drugs prescribed (prescribed drugs). The pill count calculation formula is as follows:
Pill count = Σ Drugs consumed/Σ Prescribed drugs × 100%(1)

Adherence to medications on day-1 was assessed by looking at the number of drugs and medication instructions, attendance at the previously scheduled appointment as specified on the patient’s identity card, Hospital Management Information System, and/or medication collection card. On day-1, the patient’s medication was examined and recorded. Meanwhile, adherence to medications from day-1 to day-30 was assessed by making records on the number of drugs received by the respondent up to day-30; the remaining medications were counted by the researcher on day-30. A day before the scheduled patients’ routine visits on day-30, the researcher reminded respondents via phone calls, as well as chats on WhatsApp or LINE, to bring their medications. If the respondent did not come to the scheduled appointment, the researcher would contact them by phone, as well as chat on WhatsApp or LINE. Nonetheless, if the respondent was still unable to be reached and did not attend the patients’ routine visits, he or she was moved to the drop-out category. Respondents were considered to have a high level of adherence to medications if the pill count was ≥95%, and low if it was <95% [22]. The study results were then reported to a pulmonologist and/or pharmacist responsible for the TB program and/or administrator of the TB program, to inform and ask for suggestions relating to the results.

### 2.2. Data Validity Test

A needs assessment and education planning were carried out through a preliminary study in 30 respondents who met the inclusion and exclusion criteria (excluding the research sample). Accordingly, statements or terms that were unclear to the respondents were discussed together. A difficulty index analysis was used for knowledge questions. Questions number 4 (question-related to tuberculosis disease) and 7 (question-related to anti-tuberculosis drugs administration) were considered as ‘easy’ within the difficulty index (at least 70% of respondents answer it correctly), while questions number 1, 3, 5, 6, 9, 10, 11, and 12 belong to the ‘moderate’ group (40–60% of respondents answer it correctly). Meanwhile, questions number 2 (question-related to tuberculosis disease) and 8 (question-related to anti-tuberculosis drugs administration) were considered as ‘difficult’ within the difficulty index (only 30% of respondents answer it correctly). Construct validation was done on perception. A questionnaire regarding perception consisted of 15 questions grouped into 7 domains: timeline, illness coherence, consequences, treatment control, personal control, timeline cyclical, and emotional representations. All perception questions were valid, as the product-moment correlation coefficient was above 0.3, and reliable because the Cronbach’s α test was 0.791. The Javanese (Ngoko) language on the video’s script was proofread with experts and validated by tuberculosis patients who were not respondents in the research.

### 2.3. Statistical Analysis

To compare the level of knowledge and the level of perception among the control group and intervention group, the Mann–Whitney test was used. The Wilcoxon Signed-Rank test was used to compare the level of knowledge and the level of perception per domain in each group. The level of adherence to medications in the control group and the intervention group were compared using the chi-square test, as were the relationships between respondents’ demographic factors and the level of adherence to medications in the control group and the intervention group.

### 2.4. Ethics Approval

The study was conducted according to the guidelines of the Declaration of Helsinki, and approved by the Institutional Review Board (or Ethics Committee) of Politeknik Kesehatan Kementerian Kesehatan Surabaya (the Health Research Ethics Commission of the Health Polytechnic of the Ministry of Health Surabaya), Number 025/S/KEPK/V/2017. This study acquired a research permit from Badan Kesatuan Bangsa dan Politik (the National Unity and Politics Agency) Number 072/940/424.104/SUR/RES/2018 and the Regional General Hospital (RSUD) Bangil Number 445.1/2175/424.202/2018.

## 3. Results

The characteristic demographic patients in the intervention group were similar to patients in the control group (Table 1). There was a significant difference between the knowledge level of the control group and that of the intervention group in the knowledge of tuberculosis disease, anti-tuberculosis drug administration, and anti-tuberculosis drug adverse drug events, with a *p*-value of less than 0.05 (Table 2). The consistency of improved knowledge was maintained for one month after the intervention.

Perception domains that were changed due to education through Javanese (Ngoko) videos with the CBIA method were a timeline, personal control, illness coherence, and emotional representations (*p*-value < 0.05) (Table 2). Perception domains that did not change after education was given were the consequence, treatment control, and timeline cyclical (*p*-value > 0.05).

With regards to adherence, additional education from the researcher increased the number of respondents who take 95% of their pill in the intervention group (37% increases) three times higher than in the number of the respondent in the control group (12% increases) (Table 3). There was no relationship found between respondents’ demographic factors (gender, age, level of education, and occupation) and the level of adherence to medications in the control group and the intervention group (*p*-value > 0.05) but this may be because the group size was modest.

## 4. Discussion

Many factors influenced the successful delivery of this education. First, respondents had a strong desire to recover, and this heightened their need to obtain correct information about the disease. Second, placing the video on each respondent’s mobile phone enabled patients to watch the videos again if they had forgotten. Other factors that may have affected knowledge include education level, information source, economic level, age, and occupation. An education increases adherence [23]. There is a positive relationship between knowledge level and adherence to taking anti-TB drugs [24,25,26,27]. Patients with a high level of knowledge had a greater chance of being adherent to medications compared to those having a low level of knowledge. A survey by Wandwalo and Morkve (2000) with regards to patients’ knowledge about tuberculosis revealed that only 43.9% of patients knew the cause of tuberculosis, 54.9% of patients knew how *Mycobacterium tuberculosis* bacteria are transmitted, 82% of patients assumed that tuberculosis disease could be cured, 44.3% thought that tuberculosis disease could be prevented, 50.7% of patients knew the duration of tuberculosis treatment, and 29% of patients knew the side effects of anti-TB drugs [28].

Knowledge influences perception [29,30,31,32]. Perception about illness is the patient’s experience with the disease suffered and that experience will be applied to his or her condition [33,34,35]. There is a positive correlation between perception and adherence to taking anti-TB drugs [36,37,38,39,40], and Pasek et al. (2013) found that 94% of patients with positive perception adhere to their treatment, whereas only 13% of patients with negative perception adhere to their treatment. There are 33 out of 40 tuberculosis patients (82.5%) who had a positive perception and 27 out of 40 tuberculosis patients (67.5%) had good knowledge [41].

A greater increase of respondent’s knowledge and perception in this study not only because of using a video but also because of using the local language. Language concordance will improve patient understanding, trust in the healthcare, and adherence to their treatment [42]. The implementation limitation of this study was not every healthcare had local language proficiency.

## 5. Conclusions

The use of videos with the local language, Javanese (Ngoko), as an educational tool effective increasing knowledge of tuberculosis disease, anti-TB drug administration, and anti-TB drug adverse drug events; understanding, and implicating as described in Bloom’s taxonomy; turning negative perceptions of timeline, personal control, illness coherence, and emotional representations into positives perceptions; and increasing the adherence to tuberculosis medications. 

## Figures and Tables

**Figure 1 pharmacy-09-00086-f001:**
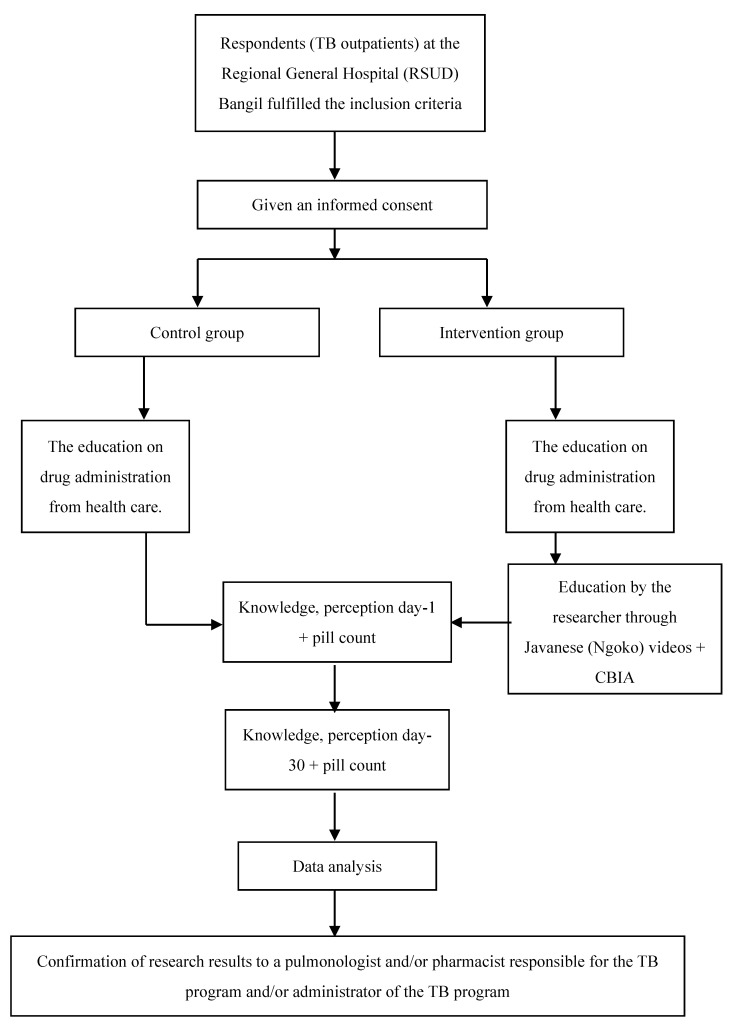
The Scheme of Research Work.

**Figure 2 pharmacy-09-00086-f002:**
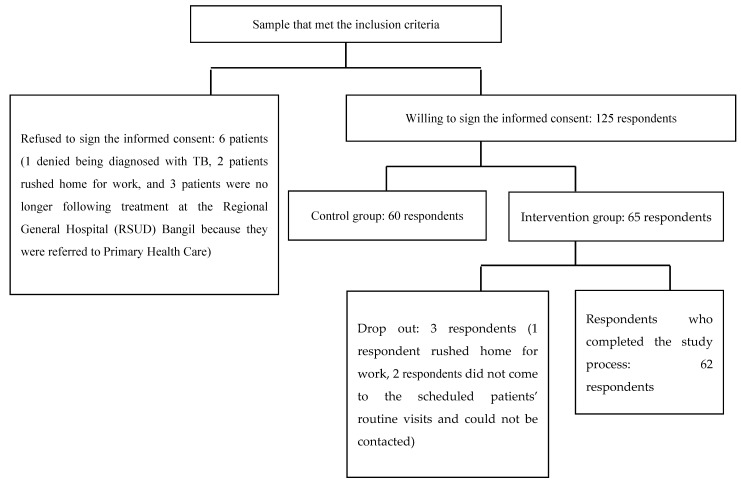
The Flow of TB Respondents’ Recruitment at the Pulmonary Outpatient Clinic in the Regional General Hospital (RSUD) Bangil.

**Table 1 pharmacy-09-00086-t001:** Baseline characteristic.

Variable	Intervention Group (*n* = 60)	Control Group (*n* = 62)	*p*-Value
Gender			0.46
Male	25 (42)	30 (48)	
Female	35 (58)	32 (52)	
Age (years old)			0.69
15 to <23	9 (15)	10 (16)	
23 to <31	16 (27)	12 (19)	
31 to <39	12 (20)	7 (11)	
39 to <47	7 (12)	12 (19)	
47 to <55	6 (10)	6 (10)	
55 to <63	7 (12)	9 (15)	
63 to <71	2 (3)	5 (8)	
≥71	1 (2)	1 (2)	
Education			0.77
Primary school	20 (33)	25 (40)	
Secondary school	13 (22)	13 (21)	
High school	20 (33)	20 (32)	
University	5 (8)	2 (3)	
Other ^1^	2 (3)	2 (3)	
Knowledge			
Lara TB ^2^	1.68	1.65	0.80
Cara ngombe OAT ^3^	1.48	1.53	0.89
Efek samping OAT ^4^	0.78	0.63	0.24
Perception			
Timeline	3.13	3.27	0.51
Consequence	6.40	6.32	0.64
Personal control	5.94	5.93	0.31
Treatment control	4.11	3.95	0.12
Illness coherence	7.00	6.82	0.30

^1^ pondok pesantren similar level with secondary school or high school. ^2^ Tuberculosis (TB) Disease. ^3^ Anti-TB Drugs (OAT, Obat Anti Tuberkulosis) administration. ^4^ Anti-TB Drugs (OAT, Obat Anti Tuberkulosis) adverse drug events.

**Table 2 pharmacy-09-00086-t002:** Average Score of Respondent’s Knowledge and Perception after intervention, day-30.

Variable	Intervention Group (*n* = 60)	Control Group (*n* = 62)	*p*-Value
Knowledge			
Lara TB ^1^	3.95	1.75	<0.001
Cara ngombe OAT ^2^	3.47	1.52	<0.001
Efek samping OAT ^3^	3.21	0.80	<0.001
Perception			
Timeline	2.56	3.30	<0.001
Consequence	6.26	6.38	0.70
Personal control	6.00	5.85	0.01
Treatment control	4.10	3.97	0.17
Illness coherence	3.00	6.70	<0.001

^1^ Tuberculosis (TB) Disease. ^2^ Anti-TB Drugs (OAT, Obat Anti Tuberkulosis) administration. ^3^ Anti-TB Drugs (OAT, Obat Anti Tuberkulosis) adverse drug events.

**Table 3 pharmacy-09-00086-t003:** Percentage of respondents’ adherence after a 30-days follow-up.

Time	Intervention Group (*n* = 60)	Control Group (*n* = 62)	*p*-Value
Pill count, day-1	58.06%	51.67%	0.48
Pill count, day-30	95.16%	63.33%	<0.001

## Data Availability

The data presented in this study are available on request from the corresponding author. The data are not publicly available due to restrictions (privacy).

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
