# Peer review of "The Effect of Javanese Language Videos with a Community Based Interactive Approach Method as an Educational Instrument for Knowledge, Perception, and Adherence amongst Tuberculosis Patients"

_pharmacy, 2021, doi:10.3390/pharmacy9020086_

Round 1

Reviewer 1 Report

Interesting and highly relevant study. Minor comments:

Abstract: Give values for results 

Page 2 lines 61-89: this is methodology and should be removed from this section

In Introduction:  to conclude section by putting forward explicit aims and objectives of the study

Methods:  there is no mention whether Ethics approval was granted

Page 3 line 113- explain acronym for DOTS

Page 3:  regarding the assessment of Drugs consumed:  not so clear how this was assessed- by asking the  patients and relying on their feedback or by asking patients to get remaining units and direct counting? 

Page 4 lines 152-153: rather than giving question numbers, it is suggested to give context of question

Page 4 lines 169-171:  to highlight that difference in knowledge is after intervention

Reviewer 2 Report

This is an interesting work presented by the authors and has a significant importance in the local Indonesian scenario

I congratulate the authors to bring the Javanese's language in the local healthcare community based interactive approach

it is very important to utilize the local languages for communications

Reviewer 3 Report

This was an interesting small study of using a video to improve knowledge and adherence to TB treatment. The paper was fairly easy to understand but there was a lot of important details missing, especially about the interventions itself. It could do with a thorough proofread (preferred native English speaker) and attention to detail. The authors could have said more about the implications for practice and scale up. 12 full name for RSUD 14 Tuberculosis does not need a capital T 18 TB not yet defined as an acronym 27 significant differences – no details on size or direction – very vague sentence 28 No conclusion in abstract – what does it mean for practice? 34 give Indonesia population – cases don’t mean much with a population denominator (like line 37) 43 – global? Careful of change from IDN to world information 64 explain more about the CBIA 65-72 – looks like methods - why is it in the Introduction? 113 define DOTS 126 How long were the videos? What content did they cover? Who made it? Is there a link to it on a website? Give details – it is the intervention! It is not clear what the control group exposure was. (122?) Fig 1 – what is involved in “education by the health personnel”? Give details. 127 What is LINE? Share it? Explain the communication platforms in general terms. Whose phone was used to send videos – e.g. a health worker’s personal phone or is it a work phone? 104 and 142 – seems to repeat information 157 What does it meant the language was checked? 178 Explain the difference – size and direction. 185 What was the starting sample size? no dropouts in control group? 187 are numbers in brackets % ? 190 knowledge aspects not clear OAT etc 192 Combine tables 1,2 and 3 so can easily see change over time. 197 Incredibly short discussion. Suggest you follow this guide: The case for structuring the discussion of scientific papers. https://www.ncbi.nlm.nih.gov/pmc/articles/PMC1115625/ 217 21 is a HUGE odds ratio 217 Nothing in the discussion about study limitations 239 Usually say about ethics approval in the methods
